# The Overlooked Floppy Eyelid Syndrome: From Diagnosis to Medical and Surgical Management

**DOI:** 10.3390/diagnostics14161828

**Published:** 2024-08-21

**Authors:** Anna Scarabosio, Pier Luigi Surico, Luca Patanè, Damiano Tambasco, Francesca Kahale, Marco Zeppieri, Pier Camillo Parodi, Marco Coassin, Antonio Di Zazzo

**Affiliations:** 1Department of Plastic Surgery, Massachusetts General Hospital, Boston, MA 02114, USA; scarabosioanna@gmail.com; 2Department of Plastic Surgery, University Hospital of Udine, 33100 Udine, Italy; piercamillo.parodi@uniud.it; 3Department of Ophthalmology, Mass Eye and Ear, Harvard Medical School, Boston, MA 02114, USA; francesca_kahale@meei.harvard.edu; 4Department of Ophthalmology, Campus Bio-Medico University Hospital, 00128 Rome, Italy; m.coassin@policlinicocampus.it (M.C.); a.dizazzo@policlinicocampus.it (A.D.Z.); 5Department of Plastic Surgery, Sapienza University of Rome, 00185 Rome, Italy; luca.patane@uniroma1.it; 6Department of Plastic Surgery, Hospital San Carlo di Nancy, 00165 Rome, Italy; damianotambasco@gmail.com; 7Department of Ophthalmology, University Hospital of Udine, 33100 Udine, Italy

**Keywords:** floppy eyelid syndrome, lid laxity, ocular surface, chronic inflammation, oculoplastics

## Abstract

Floppy Eyelid Syndrome (FES) is an underdiagnosed ocular condition characterized by the abnormal laxity of the upper eyelids, often leading to chronic eye irritation and redness. This review provides an in-depth examination of FES, covering its pathophysiology, clinical presentation, and diagnostic and therapeutic approaches. We discuss the potential etiological factors, including genetic predispositions and associations with ocular and systemic conditions such as obesity, obstructive sleep apnea, keratoconus, and glaucoma. Diagnostic strategies are outlined, emphasizing the importance of thorough clinical examinations and specific tests for an efficacious grading and assessment of FES. Management of FES ranges from conservative medical treatments to surgical interventions for more severe cases and should be driven by a comprehensive and multidisciplinary approach. Herein, we illustrate the practical aspects of diagnosing and managing this condition. This comprehensive review aims to enhance the recognition and treatment of FES, ultimately improving the quality of life for affected patients.

## 1. Introduction

Floppy Eyelid Syndrome (FES) is a relatively rare yet significant ocular condition characterized by the abnormal laxity of the upper eyelids [1]. This condition leads to the eyelids being easily everted with minimal force, causing chronic irritation, redness, and discomfort [2]. FES often affects middle-aged, overweight men, although it can occur in a broader demographic. The syndrome is frequently underdiagnosed due to its subtle presentation and the overlap of symptoms with other ocular conditions [3]. The clinical significance of FES lies in its potential to cause chronic conjunctival and corneal irritation, which can significantly impact a patient’s quality of life.

Patients commonly exhibit horizontally lax eyelids that can be easily everted with minimal lateral traction, and these eyelids lack the normal rigidity of the tarsal plate. This increased laxity often leads to chronic irritation of the papillary conjunctiva due to frequent eyelid eversion, as well as issues like lateral lid imbrication and ptosis. Consequently, individuals frequently report persistent discomfort on the ocular surface, including redness, sensitivity to light (photophobia), and a constant sensation of a foreign body in the eye [1].

The terminology used when discussing eyelid laxity and FES is pivotal. Eyelid laxity refers to the general looseness or sagging of the eyelid tissue, which can be a feature of several conditions, including FES [4]. However, FES is specifically defined by a combination of marked eyelid laxity, easy eversion of the eyelid, and associated ocular surface symptoms [5]. Distinguishing between general eyelid laxity and FES is crucial for accurate diagnosis and appropriate management.

FES was first described in 1981 by Culbertson and Ostler, who noted the characteristic findings of easily everted, floppy upper eyelids in a series of patients with chronic ocular irritation [6]. Since its initial description, understanding of the condition has evolved, recognizing its association with systemic conditions such as obstructive sleep apnea and obesity [7,8,9]. Over the years, FES has garnered attention not only from ophthalmologists but also from other medical specialists due to its systemic associations and impact on overall health [10].

FES is considered an uncommon condition, but its exact prevalence is difficult to determine due to underdiagnosis and misdiagnosis. Epidemiological studies suggest that FES is more prevalent among middle-aged, overweight men, though it can affect individuals of any gender and age group [11]. The condition is often associated with obesity, with a significant number of affected individuals having a higher body mass index (BMI) [9,12]. Recent studies have also indicated a strong correlation between FES and obstructive sleep apnea (OSA) [13]. Approximately 25–60% of patients with OSA are reported to have FES, indicating that the two conditions may share common pathophysiological mechanisms [14]. Additionally, FES has been observed in patients with other systemic conditions such as diabetes mellitus, hypertension, and hyperglycinemia, although these associations are less well defined [15,16,17].

The condition is more frequently diagnosed in ophthalmic settings, where patients present with chronic ocular surface irritation and seek specialist care. However, given the subtlety of its symptoms and overlap with other ocular conditions, many cases of FES may remain undetected in the general population.

## 2. Materials and Methods

This review conducted a comprehensive literature search using PubMed and Reference Citation Analysis (RCA). PubMed, managed by the National Library of Medicine, was selected for its extensive repository of peer-reviewed biomedical literature. The search strategy included terms related to “Floppy Eyelid Syndrome” in combination with “lid laxity”, “pathogenesis”, “obstructive sleep apnea”, “OSA”, “treatment”, “diagnosis”, “ocular surface”, “inflammation”, “surgical treatment”, “oculoplastics”, and “therapy”. Boolean operators (AND, OR, NOT) were applied to ensure a thorough and relevant search. The review focused on English-language articles to maintain accessibility. Titles and abstracts were manually screened, and full texts of pertinent articles were reviewed to extract information on clinical features, treatment options, outcomes, and related challenges. Additionally, manual searches of reference lists and citation tracking were conducted to supplement the electronic search. This methodology was designed to capture a comprehensive overview of Floppy Eyelid Syndrome, encompassing current knowledge and advancements.

## 3. Pathophysiology

As already mentioned, FES is characterized by the excessive laxity of the upper eyelid, leading to its eversion or inversion, especially during sleep. This laxity causes chronic irritation and can result in significant ocular surface disease. The pathophysiology of Floppy Eyelid Syndrome (FES) involves several interrelated factors. While the exact mechanism behind FES has yet to be fully delineated, current evidence suggests that it likely results from a combination of influences. These include mechanical trauma, oxidative damage due to hypoxia and reperfusion, associations with systemic conditions, and potential genetic factors. The hallmark of FES is the abnormal looseness of the upper eyelid, which can be attributed to alterations in the structure and function of elastin and collagen within the eyelid connective tissue. Elastin, in particular, is crucial for maintaining the elasticity and firmness of tissues, and its dysfunction plays a key role in the development of the disease. In 1994, Netland et al. demonstrated that elastin fibers are degraded or abnormally organized, leading to the eyelid’s increased compliance and propensity to evert [18]. Moreover, in 2005, Schlötzer-Schrehardt et al. indicated that upregulation of elastolytic enzymes participates in elastic fiber degradation and subsequent tarsal laxity and eyelash ptosis in FES [19,20].

### 3.1. Mechanical Trauma

Patients with FES often sleep in a manner that exerts pressure on the eyelids, such as face-down or side-sleeping positions. The combination of mechanical pressure and the inherent laxity of the eyelid can cause the eyelid to invert or evert during sleep, exposing the ocular surface to the external environment. This repeated mechanical trauma contributes to chronic irritation, inflammation, and sometimes secondary infections [21,22,23].

### 3.2. Oxidative Damage from Hypoxia and Reperfusion

Ischemia has been proposed as a significant pathogenetic factor in Floppy Eyelid Syndrome (FES), contributing to the structural degradation and functional impairment of the eyelid. As observed in the study by Culbertson and Tseng, local pressure-induced ischemia plays a critical role in the degenerative changes seen in the tarsal plate. Patients with FES often sleep face down, exerting continuous pressure on the eyelid, which can lead to localized ischemia [24]. This ischemia, compounded by systemic conditions such as hypoventilation, results in chronic tissue hypoxia [25]. Hypoxia not only stems from obstructive sleep apnea (OSA) but also from the repetitive mechanical pressure on the eyelids during sleep [26]. The ensuing reperfusion injury, characterized by oxidative stress upon reoxygenation, exacerbates the damage by promoting inflammation and the activation of matrix metalloproteinases (MMPs) [27,28]. These enzymes degrade elastin and collagen fibers, leading to the hallmark eyelid laxity. Furthermore, the study highlighted that both clinical and subclinical corneal disorders, such as punctate epithelial keratopathy and keratoconus, are common in FES patients, suggesting that corneal hypoxia may also play a role in disease progression [24]. Thus, ischemia-induced hypoxia and subsequent reperfusion injury are pivotal in understanding the pathogenesis of FES, emphasizing the need for treatments that address both mechanical and systemic contributors to the condition. In addition to the localized pressure-induced ischemia, patients with OSA experience intermittent systemic hypoxia during episodes of apnea and hypopnea. This combination of factors gradually compromises the integrity of the connective tissue, resulting in the characteristic laxity of the eyelids.

### 3.3. Association with Systemic Conditions

FES is often associated with systemic conditions such as obesity, OSA, and connective tissue disorders like Ehlers–Danlos syndrome. The exact mechanism linking these conditions to FES is not fully understood, but it is believed that increased intraocular pressure during apneic episodes in OSA and the altered connective tissue structure in obesity and connective tissue disorders play a significant role [9,29].

### 3.4. Genetic Factors

Current evidence indicates a genetic predisposition to FES, as evidenced by cases in which individuals lack other etiopathogenetic factors such as mechanical trauma or obstructive sleep apnea syndrome (OSAS). Variations in genes responsible for elastin and collagen production may predispose individuals to develop the syndrome. Ehlers–Danlos syndrome, a collagenopathy characterized by inadequate production of type V collagen, has been frequently studied and associated with FES [30,31]. A possible predisposition in individuals with Down syndrome was described. A case was reported of a two-year-old child with significant upper eyelid laxity and a tendency for nocturnal eversion, without other comorbidities. There is debate regarding whether this is due to a genetic cause (such as canthal alteration or tarsal pathology) or if these children tend to rub their eyes more frequently. The latter hypothesis seems less likely given the early onset of symptoms [32,33,34]. Additionally, Caccavale et al. reported a possible association between neurofibromatosis type 1 and FES in a case report describing an 11-year-old child [35].

### 3.5. Histological Characteristics

In FES, significant anatomical and histological changes contribute to the characteristic laxity and dysfunction of the eyelids. Anatomically, the upper eyelid exhibits marked laxity, allowing it to easily evert or invert, particularly during sleep. This excessive laxity is primarily due to alterations in the tarsal plate. Histological examination reveals degeneration and fragmentation of elastin fibers, which are crucial for maintaining tissue elasticity [18,36,37]. Additionally, collagen fibers, which provide tensile strength, may show disorganization and reduced density. These changes lead to the compromised structural integrity of the eyelid. Inflammatory cell infiltration, particularly with lymphocytes and macrophages, is often observed, indicating a chronic inflammatory state. This inflammation further exacerbates tissue degradation and contributes to the clinical manifestations of FES, including eyelid malposition and ocular surface disease. Understanding these anatomical and histological alterations is essential for comprehending the pathophysiology of FES and guiding effective treatment strategies.

## 4. Clinical Presentation

FES presents distinctive clinical characteristics that are evident in both signs and symptoms. Patients typically display horizontally lax eyelids that can be easily everted with minimal lateral traction, lacking the normal rigidity of the tarsal plate [15,37,38]. This laxity often leads to chronic irritation of the papillary conjunctiva due to frequent eyelid eversion, along with lateral lid imbrication and ptosis [39,40].

Symptomatically, individuals commonly report persistent ocular surface discomfort, including redness, sensitivity to light (photophobia), and a constant foreign body sensation [41,42]. They also frequently experience mucoid discharge and they can present with filamentary keratitis [43]. These symptoms tend to persist over the long term, with exacerbations often noted in the morning upon waking, typically affecting the side on which they sleep, although FES has been shown to be bilateral in a significant percentage of patients [15].

FES is also associated with several other eyelid conditions, such as blepharitis, ectropion, entropion, blepharochalasis, and additional lid ptosis [44,45,46,47]. Notably, lash ptosis strongly indicates FES and should prompt further clinical evaluation [20]. Many patients with FES present with corneal complications, including exposure to keratopathy and, in some instances, keratoconus [10,13,48,49,50].

In FES, the chronic exposure of the corneal surface due to eyelid laxity significantly increases the risk of severe corneal complications [51]. Persistent exposure can lead to the development of corneal ulcers, which are concerning because they can progress to perforation, a condition that threatens vision [52]. Additionally, the compromised corneal surface is highly susceptible to infections, which can further exacerbate damage and compromise visual function [53]. Prompt and effective management is crucial to address these risks and prevent permanent damage [54].

## 5. Comorbidities

### 5.1. Obstructive Sleep Apnea

Given that FES may be a manifestation of an underlying systemic condition, a thorough evaluation for both ocular and systemic diseases is recommended. Approximately 25–60% of patients with OSA are reported to have FES, indicating that the two conditions may share common pathophysiological mechanisms [14]. The strong correlation between FES and Obstructive Sleep Apnea (OSA) means that if OSA is present, it should prompt further investigation for FES [4,5,8,14,55,56,57]. Conversely, all patients diagnosed with FES should be screened for OSA. This includes gathering a detailed history of symptoms such as snoring, family history of snoring, daytime sleepiness, dizziness, nocturnal awakenings, headaches, and uncontrolled hypertension [58,59]. Referral to a sleep disorder specialist for polysomnography may be warranted. Additionally, due to the genetic components associated with OSA, older children of patients with suspected FES and undiagnosed OSA should also undergo screening for OSA [60,61].

In summary, the clinical presentation of FES includes characteristic eyelid laxity and associated ocular surface symptoms, often accompanied by specific ocular findings and demographic correlations. Recognizing these features and understanding the potential systemic connections is crucial for accurate diagnosis and effective management of the condition.

### 5.2. Keratoconus

FES is frequently associated with keratoconus, a progressive thinning and cone-shaped deformation of the cornea [49,50,62]. Both conditions share a common etiological factor: eye rubbing [4,63]. The prevalence of keratoconus in patients with FES is not well defined in the literature, but some studies suggest that corneal disorders, including keratoconus, are relatively common among FES patients. Estimates vary, but it has been reported that approximately 10–20% of patients with FES may also have keratoconus [64]. In patients with FES, the severity of keratoconus is often greater on the side on which they sleep due to eyelid eversion during sleep, leading to increased mechanical exposure [23]. However, the connection between FES and keratoconus extends beyond mechanical trauma. Research indicates that there is an increased expression of matrix metalloproteinases in individuals with these conditions, leading to enhanced tissue and matrix degradation [19,65]. This enzymatic activity further weakens the corneal structure, contributing to the progression of keratoconus and underscoring the multifaceted nature of the association between these two disorders. Additionally, FES has also been reported as a potential risk factor for cross-linking failure in patients with keratoconus [66].

### 5.3. Glaucoma

FES is closely associated with an increased risk of developing glaucoma, particularly primary open-angle glaucoma (POAG) and normal-tension glaucoma (NTG), due to its strong connection with obstructive sleep apnea (OSA) [67,68,69,70,71,72]. Studies have shown that patients with OSA are at a higher risk of developing both POAG and NTG, and FES serves as a predictive factor for glaucoma in these individuals [73]. In a recent study, patients with OSA have around a 40% higher likelihood of developing glaucoma compared to normal individuals [64,74]. Given this significant correlation, it is crucial to screen for glaucoma in patients with FES. Regular monitoring of intraocular pressure (IOP) and comprehensive optic nerve examinations are essential in these patients to detect early signs of glaucoma and initiate timely management [75,76]. Early detection and intervention can help prevent the progression of glaucoma and preserve vision in this high-risk population.

## 6. Diagnosis

Diagnosing Floppy Eyelid Syndrome (FES) involves a comprehensive assessment that integrates clinical examination and specific diagnostic tests (Table 1).

Clinical examination begins with a detailed evaluation of eyelid function. This includes assessing the ease with which the upper eyelid can be everted and noting any spontaneous eversion during examination [3,6,51,77]. Patients with FES typically exhibit significant eyelid laxity, allowing the eyelid to be easily turned inside out with minimal pressure or during sleep, leading to chronic ocular irritation. The examination also includes evaluating for signs of chronic inflammation such as papillary conjunctivitis, conjunctival hyperemia, and corneal epithelial changes.

Diagnostic tests play a crucial role in confirming the diagnosis and assessing the severity of FES. The snap-back test is commonly used to objectively evaluate eyelid laxity [85,86]. By gently pulling the eyelid away from the globe and observing the speed and extent of its return, clinicians can assess the degree of laxity. In FES, the eyelid often returns slowly and may not fully revert to its normal position due to increased laxity.

Fluorescein staining is employed to visualize corneal epithelial defects caused by chronic exposure due to eyelid laxity [87,88]. This diagnostic tool helps quantify the extent of ocular surface compromise and guides management strategies.

Other tests further aid in confirming the diagnosis and evaluating ocular surface health. The Schirmer test measures tear production by placing strips of filter paper inside the lower eyelid to assess the quantity of tears produced over a specified time. It helps in identifying aqueous tear deficiency, which may coexist with FES and exacerbate ocular surface symptoms [89].

Tear Break-up Time (TBUT) measures the interval between a blink and the appearance of dry spots or discontinuity in the tear film on the cornea. A shortened TBUT suggests instability of the tear film, contributing to symptoms of dryness and irritation in patients with FES [91].

Meibomian gland examination, through techniques like meibography or assessment of gland function, evaluates meibomian gland dysfunction (MGD) commonly associated with FES [89,92]. Dysfunction of these glands can lead to poor lipid layer quality in the tear film, exacerbating ocular surface symptoms.

These diagnostic tools provide objective measures of tear film stability, tear production, and meibomian gland function, complementing the clinical evaluation of FES. They help guide management strategies aimed at alleviating symptoms, improving ocular surface health, and enhancing patient comfort.

Clinically grading the severity of FES involves assessing the visibility of the upper tarsal conjunctiva [1]. The test requires the clinician to lift the upper eyelid with their thumb while the patient looks downward. This grading system, described in Table 2, helps categorize the extent of eyelid laxity and guides treatment decisions.

Differential diagnosis is essential to distinguish FES from other conditions presenting with eyelid laxity and ocular surface irritation, such as blepharochalasis and dermatochalasis [46,96]. These conditions may share some clinical features but lack the pronounced eyelid findings characteristic of FES.

## 7. Conservative Management

In managing ocular surface disease associated with FES, there is no single preferred method. Instead, a combination of strategies is used to address all contributing factors, including dryness, chronic inflammation, eyelid laxity, and meibomian gland dysfunction or lid inflammation. While conservative treatment seeks to manage these elements comprehensively, it can be complemented by interventional surgical approaches. These combined methods can work together to achieve improved overall outcomes.

### 7.1. Ocular Surface Lubrication

Maintaining lubrication and moisture of the ocular surface is essential for relieving the symptoms of Floppy Eyelid Syndrome (FES). This approach also contributes to reducing inflammation and restoring homeostasis of the ocular surface [78,79]. It is important for non-ophthalmologist caregivers to recognize this need and adopt a comprehensive management strategy for the disease, which is often associated with systemic conditions such as OSA [42]. Nighttime use of ointments or gels, which stay on the eye longer than eyedrops, provides better protection for the cornea and ocular surface and prevents worrisome consequences from exposure. It is crucial to use preservative-free products as preservatives can paradoxically cause ocular surface toxicity and worsen inflammation [90]. It is also mandatory to prevent patients from using topical anesthetic eye drops that provide symptomatic relief but significantly increase the risk of vision loss due to corneal melting [97]. The period of using lubricant eyedrops is typically long-term or even indefinitely, as they provide symptomatic relief and help manage the dryness associated with chronic ocular surface conditions. Lubricant eyedrops are considered a supportive treatment rather than a definitive cure, given the chronic nature of the pathology.

### 7.2. Eyelid Taping

Before considering surgical intervention, eyelid taping combined with the application of ointment during sleep has proven effective in preventing exposure [80]. This method is useful not only in FES but also in other conditions like thyroid eye disease, where there is incomplete or ineffective eyelid closure during the night [53,98].

### 7.3. Meibomian Gland Dysfunction

In patients with concomitant blepharitis and meibomian gland disease, it is crucial to optimize these conditions. This involves consistent daily application of heat, eyelid hygiene, antibiotic ointment (such as azithromycin), and, in severe cases, oral low-dose tetracycline or azytromycin to reduce lid inflammation. Antibiotics like tetracyclines are used in MGD primarily for their anti-inflammatory properties. While tetracyclines are antibiotics, their effectiveness in MGD comes from their ability to reduce inflammation and modulate the immune response in the eyelid. Tetracyclines can help improve meibomian gland function, decrease the secretion of pro-inflammatory cytokines, and reduce the associated symptoms of dry eye and discomfort. They are often prescribed in lower doses for their long-term anti-inflammatory benefits rather than their antibacterial effects [93,94,95].

### 7.4. Topical Anti-Inflammatory Therapy

In cases of acute or exacerbated ocular surface inflammation, short courses of anti-inflammatory topical therapy may be considered. Mild corticosteroids such as loteprednol or fluorometholone can help reduce inflammation with a minimal effect on intraocular pressure [81,82]. It is recommended to avoid non-steroidal anti-inflammatory eyedrops as they can increase the long-term risk of corneal melting, a vision-threatening complication that should be avoided [83,84]. Moreover, it is mandatory to monitor intraocular pressure when using steroids, particularly in patients with OSAS who already have a higher risk of developing glaucoma [73,99].

### 7.5. Continuous Positive Airway Pressure (CPAP)

Managing OSA comprehensively is also critical. Studies have shown that patients can benefit from CPAP therapy, not only for the ocular aspects but also for improving snoring and other manifestations of the disease spectrum [100,101]. Although CPAP can initially irritate the ocular surface, it has the potential to reduce chronic inflammation and improve ocular symptoms over time [55].

### 7.6. Oculoplastic Evaluation

Early referral to an oculoplastic specialist is important for a proper evaluation of the patient, which helps in making informed decisions about whether a conservative or more interventional approach is needed. This comprehensive approach ensures that all aspects of FES and its associated conditions are addressed effectively.

## 8. Surgical Treatment

Floppy Eyelid Syndrome (FES) is characterized by excessive upper eyelid laxity, which may be surgically addressed. While conservative treatments are often first-line, surgical intervention may be necessary for severe or refractory cases. Surgical techniques aim to restore eyelid function by tightening and supporting the eyelid structure. Several techniques have been described.

### 8.1. Lateral Tarsal Strip Procedure

The lateral tarsal strip procedure is one of the most common and effective surgical treatments for FES (Figure 1). This technique involves shortening and reinforcing the lateral portion of the tarsal plate, which provides structural support to the eyelid. The procedure begins with an incision at the lateral canthus, followed by the dissection of the lateral tarsal plate. The surgeon then excises a strip of the tarsal plate to the desired length, ensuring sufficient tension to correct the eyelid laxity. The remaining tarsal plate is then reattached to the periosteum of the lateral orbital rim or the lateral canthal tendon using non-absorbable sutures [102]. This repositioning restores the eyelid’s normal tension and contour, reducing symptoms and preventing further eversion. The lateral tarsal strip procedure is highly effective, with a success rate exceeding 90%, and is often considered the gold standard for FES surgical treatment [39].

### 8.2. Medial Canthoplasty

Medial canthoplasty is another surgical option for treating FES, particularly in cases where medial eyelid laxity is more pronounced. This procedure involves tightening the medial canthal tendon and repositioning the medial aspect of the eyelid. The surgeon begins by making an incision near the medial canthus and dissecting the underlying tissues to expose the medial canthal tendon. The tendon is then shortened and reattached to the periosteum of the medial orbital rim using non-absorbable sutures. This technique effectively tightens the medial eyelid, reducing laxity and improving overall eyelid function [103,104,105]. Medial canthoplasty can be performed alone or in combination with the lateral tarsal strip procedure for comprehensive eyelid tightening [106].

### 8.3. Full-Thickness Wedge Resection

Full-thickness wedge excision (FTWE) procedures are commonly used for malignancies that require excision and have associated eyelid laxity. This technique involves removing a full-thickness wedge of tissue from the affected eyelid to reduce laxity and improve eyelid tension [107]. The procedure starts with marking a vertical area of resection, usually in the central or lateral portion of the eyelid. After excising the wedge, the remaining edges are sutured together, resulting in a tighter, more stable eyelid. This technique can be particularly useful for patients with localized areas of significant laxity [108,109].

### 8.4. Upper Eyelid Blepharoplasty

Primarily used for cosmetic purposes, upper eyelid blepharoplasty can also benefit FES patients by removing excess skin and tissue that contribute to eyelid laxity. The procedure involves making an incision along the natural crease of the upper eyelid, excising the redundant skin and muscle, and closing the incision with fine sutures. This technique not only improves the aesthetic appearance but also enhances eyelid function by reducing the excess weight and laxity. Usually, it cannot be the only procedure performed but needs to be associated to a specific lid tightening [46].

### 8.5. Canthal Suspension Procedures

These procedures involve suspending the eyelid to the orbital rim or other stable structures to provide additional support. Techniques such as lateral canthal sling or suture suspension can be used to reposition and stabilize the eyelid, reducing laxity and preventing eversion [110]. These procedures are often used in conjunction with other surgical techniques to achieve optimal results [111].

Surgical options for Floppy Eyelid Syndrome (FES) are diverse and tailored to address the specific needs of each patient. The lateral tarsal strip procedure is the most common and effective method, providing significant improvement in eyelid function and symptom relief. Medial canthoplasty and other techniques such as full-thickness wedge resection, upper eyelid blepharoplasty, and canthal suspension procedures offer additional strategies to manage eyelid laxity. Selecting the appropriate surgical approach depends on the severity and location of the eyelid laxity, as well as the patient’s overall health and preferences. With careful assessment and skilled surgical intervention, patients with FES can achieve substantial symptomatic relief and improved quality of life.

## 9. Conclusions

In conclusion, increasing knowledge and awareness of Floppy Eyelid Syndrome (FES) is essential for improving patient management and overall health. FES can manifest as part of the spectrum of systemic conditions such as OSA, affecting not only ocular health but also significantly impacting patients’ lives. Appropriate diagnosis and screening for comorbidities can lead to optimal management through a multidisciplinary approach. Cooperation between ophthalmologists, internal medicine specialists, and oculoplastic surgeons ensures comprehensive care, addressing both ocular and systemic aspects of FES and preventing complications. Enhanced understanding among healthcare providers will ultimately lead to better patient outcomes and improved quality of life.

## Figures and Tables

**Figure 1 diagnostics-14-01828-f001:**
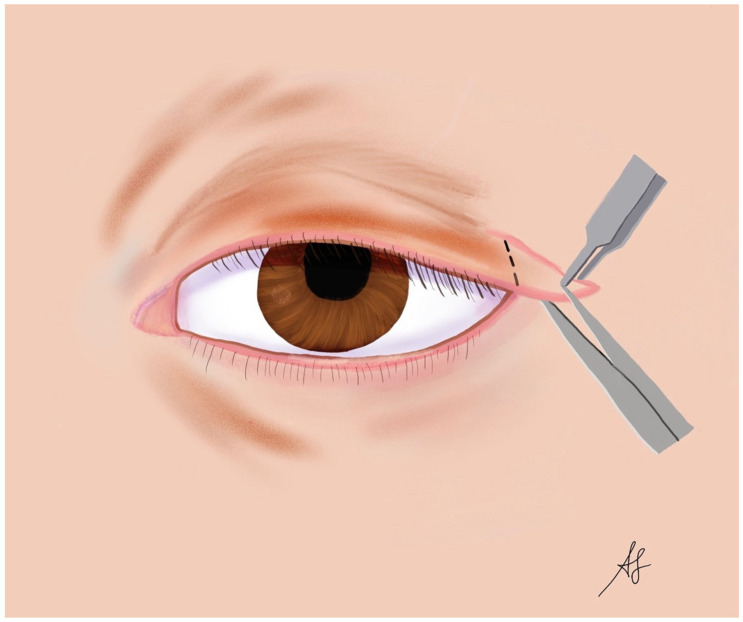
Lateral tarsal strip surgery in floppy eyelid syndrome (FES): a graphical representation (designed and created by AS).

**Table 1 diagnostics-14-01828-t001:** Clinical examination for diagnosing floppy eyelid syndrome (FES), and respective therapeutic approaches.

Aspect	Diagnostic Approach	Therapeutic Approach
**Eyelid Function**	Assessment of the ease of upper eyelid eversion and observation of any spontaneous eversion during examination. FES patients typically exhibit significant laxity [3,6,51,77].	Nighttime use of preservative-free ointments or gels for better protection. Eyelid taping combined with the application of ointment during sleep to prevent exposure. Early referral to an oculoplastic specialist for further evaluation [78,79,80].
**Chronic Inflammation**	Evaluation for signs such as papillary conjunctivitis, conjunctival hyperemia, and corneal epithelial changes [2].	Maintaining ocular surface lubrication with preservative-free products. Use of short courses of mild corticosteroids like loteprednol or fluorometholone for acute inflammation. Avoidance of non-steroidal anti-inflammatory eyedrops for risk of corneal melting [81,82,83,84].
**Snap-back Test**	Gentle pulling of the eyelid away from the globe and observing its return speed and extent. In FES, the eyelid returns slowly and may not fully revert [85,86].	Similar to eyelid function management. Addressing underlying systemic conditions like Obstructive Sleep Apnea (OSA). Continuous Positive Airway Pressure (CPAP) therapy to improve chronic inflammation and ocular symptoms [42].
**Fluorescein Staining**	Visualization of corneal epithelial defects caused by chronic exposure due to eyelid laxity, aiding in quantifying ocular surface compromise [87,88].	Use of preservative-free lubricating drops, gels, or ointments to protect the cornea. Avoiding non-steroidal anti-inflammatory eyedrops to prevent corneal melting [83,84].
**Schirmer Test**	Measurement of tear production using filter paper strips placed inside the lower eyelid to identify aqueous tear deficiency [89].	Enhancing tear film volume and stability with preservative-free lubricants and managing chronic inflammation [90].
**Tear Break-up Time (TBUT)**	Measurement of the interval between a blink and the appearance of dry spots on the cornea, indicating tear film instability [91].	Manage low tear film and, importantly, meibomian gland dysfunction and blepharitis, to improve tear film quality. Additionally, oral low-dose tetracycline for severe cases to reduce lid inflammation [89,92,93,94,95].
**Meibomian Gland Examination**	Evaluation of meibomian gland dysfunction (MGD) through techniques like meibography or gland function assessment [89,92].	Consistent daily application of heat, eyelid hygiene, and antibiotic ointments to manage blepharitis. Oral low-dose tetracycline in severe cases. Addressing systemic conditions like OSA to reduce ocular symptoms. Early referral to a specialist if necessary [93,94,95].

**Table 2 diagnostics-14-01828-t002:** Grading of Floppy Eyelid Syndrome (FES) [1].

Grade	Description
**0**	No floppy eyelid; no visible tarsal conjunctiva.
**1**	Mild severity; less than one-third of the upper tarsal conjunctiva is visible.
**2**	Moderate severity; between one-third and one-half of the upper tarsal conjunctiva is visible.
**3**	Severe cases; more than half of the tarsal conjunctiva is visible.

## Data Availability

No new data were created or analyzed in this study. Data sharing is not applicable to this article.

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
