# Peer review of "The Overlooked Floppy Eyelid Syndrome: From Diagnosis to Medical and Surgical Management"

_diagnostics, 2024, doi:10.3390/diagnostics14161828_

Round 1

Reviewer 1 Report

Comments and Suggestions for Authors

Dear author,

the manuscript becomes more valuable if consider the following:

1.  add details on symptoms of Floppy Eyelid Syndrome in the introduction section

2.  The pathogenesis of the Floppy Eyelid Syndrome is unclear and needs a more precise cause, especially in hypoxia condition 

3.  The terminology used when discussing eyelid laxity and FES is pivotal, what is the meaning of pivotal? please discuss this in the section on terminology

4. in the comorbidity section: add the participation percent of each co-morbidity, for example in the glaucoma section, what is the percentage of patients with glaucoma who can develop Floppy Eyelid Syndrome?

5.  in table 1: diagnostic and therapeutic approaches need references for each one.

6.  why in Meibomian Gland Examination use antibiotics, please discuss this.

7.  in the management of Chronic Inflammation, why advise patient to avoid NSAIDs?

8. add references to Table 2

9. in the Conservative Management, section 7.1 Ocular surface lubrication, what is the period of using lubricant? 

10.   in Conservative Management, what is the preferred method?

With regards,

Comments on the Quality of English Language

minor editing is required.

Author Response

Dear author,

the manuscript becomes more valuable if consider the following:

  1. add details on symptoms of Floppy Eyelid Syndrome in the introduction section

We thank the reviewer for pointing this out. We added a brief presentation of FES symptoms in the introduction.

“Patients commonly exhibit horizontally lax eyelids that can be easily everted with minimal lateral traction, and these eyelids lack the normal rigidity of the tarsal plate. This increased laxity often leads to chronic irritation of the papillary conjunctiva due to frequent eyelid eversion, as well as issues like lateral lid imbrication and ptosis. Consequently, individuals frequently report persistent discomfort on the ocular surface, including redness, sensitivity to light (photophobia), and a constant sensation of a foreign body in the eye. [1] “

 However, the clinical manifestations has been detailed in the specific section 4: Clinical Presentation.

  1. The pathogenesis of the Floppy Eyelid Syndrome is unclear and needs a more precise cause, especially in hypoxia condition.

We thank the reviewer for the comment. While the exact mechanism behind Floppy Eyelid Syndrome (FES) has yet to be fully delineated, current evidence suggests that it likely involves a combination of factors. These include mechanical trauma, oxidative damage due to hypoxia and reperfusion, associations with systemic conditions, and potential genetic factors. Together, these elements contribute to alterations in the structure and function of elastin and collagen within the eyelid connective tissue, which are key to the development of FES, as mentioned. Further research is certainly needed to better understand these complex interactions. Moreover, in the “Oxidative damage from hypoxia and reperfusion” we aim to underline the role of ischemia, particularly from local pressure during sleep, as key factor in the pathogenesis of FES, leading to structural degradation of the eyelid. This ischemia, compounded by systemic conditions like OSA, results in chronic tissue hypoxia and reperfusion injury, which exacerbates eyelid laxity through oxidative stress and the activation of matrix metalloproteinases. The combination of mechanical pressure and systemic hypoxia gradually compromises connective tissue integrity, potentially contributing to the characteristic eyelid laxity in FES.

We reviewed the introduction to the Section 3 “Pathophysiology” where we delineate the potential mechanisms inducing FES (including hypoxia condition).

“The pathophysiology of Floppy Eyelid Syndrome (FES) involves several interrelated factors. While the exact mechanism behind FES has yet to be fully delineated, current evidence suggests that it likely results from a combination of influences. These include mechanical trauma, oxidative damage due to hypoxia and reperfusion, associations with systemic conditions, and potential genetic factors. The hallmark of FES is the abnormal looseness of the upper eyelid, which can be attributed to alterations in the structure and function of elastin and collagen within the eyelid connective tissue. Elastin, in particular, is crucial for maintaining the elasticity and firmness of tissues, and its dysfunction plays a key role in the development of the disease.”

  1. The terminology used when discussing eyelid laxity and FES is pivotal, what is the meaning of pivotal? please discuss this in the section on terminology

We thank the reviewer for the comment. As we have already addressed the terminology in the section, it's important to clarify that "pivotal" refers to something of central significance. In the context of eyelid laxity FES, precise terminology is crucial for accurately conveying the importance and nuances of the condition. This ensures that key concepts and findings are communicated clearly and effectively.

  1. in the comorbidity section: add the participation percent of each co-morbidity, for example in the glaucoma section, what is the percentage of patients with glaucoma who can develop Floppy Eyelid Syndrome?

We thank the reviewer for pointing this out. While the percentages provided are estimates and further large cohort studies are needed to confirm these numbers, we have addressed the reviewer's question in the manuscript. Specifically, for OSA, we report that approximately 25-60% of patients with OSA are also reported to have FES, suggesting a potential shared pathophysiology [14]. Regarding keratoconus, although the prevalence in FES patients is not well-defined, estimates suggest that around 10-20% of FES patients may have keratoconus [64]. For glaucoma, a recent study indicates that patients with OSA have approximately a 40% higher likelihood of developing glaucoma compared to the general population [64,74].

  1. in table 1: diagnostic and therapeutic approaches need references for each one.

We thank the reviewer for the comment. We added the references accordingly.

  1. why in Meibomian Gland Examination use antibiotics, please discuss this.

We thank the reviewer for pointing this out. We clarified the rationale in section 7.3, as follows:

“Antibiotics like tetracyclines are used in meibomian gland dysfunction (MGD) primarily for their anti-inflammatory properties. While tetracyclines are antibiotics, their effectiveness in MGD comes from their ability to reduce inflammation and modulate the immune response in the eyelid. Tetracyclines can help improve meibomian gland function, decrease the secretion of pro-inflammatory cytokines, and reduce the associated symptoms of dry eye and discomfort. They are often prescribed in lower doses for their long-term anti-inflammatory benefits rather than their antibacterial effects.”

  1. in the management of Chronic Inflammation, why advise patient to avoid NSAIDs?

We thank the reviewer for the comment. As mentioned in section 7.4, in the management of chronic inflammation, it is advisable to avoid non-steroidal anti-inflammatory (NSAID) eyedrops because they can increase the long-term risk of corneal melting, a serious and vision-threatening complication [83,84].

  1. add references to Table 2

We added the reference.

  1. in the Conservative Management, section 7.1 Ocular surface lubrication, what is the period of using lubricant? 

We thank the reviewer for the comment. We clarified in section 7.1 on ocular surface lubrication, as follows:

“The period of using lubricant eyedrops is typically long-term or even indefinitely, as they provide symptomatic relief and help manage the dryness associated with chronic ocular surface conditions. Lubricant eyedrops are considered a supportive treatment rather than a definitive cure, given the chronic nature of the pathology.”

 As stated in the manuscript, there are generally no side effects associated with the use of preservative-free lubricants, making them a safe option for ongoing use.

  1. in Conservative Management, what is the preferred method?

We thank the reviewer for giving us the opportunity to better clarify this concept. To address the comment, we added the following paragraph in section 7:

“In managing chronic ocular conditions, there is no single preferred method. Instead, a combination of strategies is used to address all contributing factors, including inflammation, dryness, chronic inflammation, eyelid laxity, and meibomian gland dysfunction or lid inflammation. While conservative treatment seeks to manage these elements comprehensively, it can be complemented by interventional surgical approaches. These combined methods can work together to achieve improved overall outcomes.”

Reviewer 2 Report

Comments and Suggestions for Authors

this is a very good review of the subject and might be a good tool for every doctor that deals with Floppy Eyelid Syndrome.  

Well Done

Author Response

this is a very good review of the subject and might be a good tool for every doctor that deals with Floppy Eyelid Syndrome.  

Well Done

We thank the reviewer for the positive feedback. We are glad to hear that the reviewer found our article valuable. We are hopeful that this manuscript could serve as a useful resource for clinicians managing Floppy Eyelid Syndrome.